# From Genes to Molecules, Secondary Metabolism in *Botrytis cinerea*: New Insights into Anamorphic and Teleomorphic Stages

**DOI:** 10.3390/plants12030553

**Published:** 2023-01-26

**Authors:** Haroldo da Silva Ripardo-Filho, Víctor Coca Ruíz, Ivonne Suárez, Javier Moraga, Josefina Aleu, Isidro G. Collado

**Affiliations:** 1Departamento de Química Orgánica, Facultad de Ciencias, Universidad de Cádiz, Puerto Real, 11510 Cádiz, Spain; 2Departamento de Biomedicina, Biotecnología y Salud Pública, Laboratorio de Microbiología, Facultad de Ciencias del Mar y Ambientales, Universidad de Cádiz, Puerto Real, 11510 Cádiz, Spain; 3Instituto de Investigación en Biomoléculas (INBIO), Universidad de Cádiz, Puerto Real, 11510 Cádiz, Spain

**Keywords:** *Botrytis cinerea*, *Botryotinia fuckeliana*, secondary metabolism, metabolites, sesquiterpenes, diterpenes, polyketides

## Abstract

The ascomycete *Botrytis cinerea* Pers. Fr., classified within the family Sclerotiniaceae, is the agent that causes grey mould disease which infects at least 1400 plant species, including crops of economic importance such as grapes and strawberries. The life cycle of *B. cinerea* consists of two phases: asexual (anamorph, *Botrytis cinerea* Pers. Fr.) and sexual (teleomorph, *Botryotinia fuckeliana* (de Bary) Wetzel). During the XVI International Symposium dedicated to the *Botrytis* fungus, which was held in Bari in June 2013, the scientific community unanimously decided to assign the most widely used name of the asexual form, *Botrytis*, to this genus of fungi. However, in the literature, we continue to find articles referring to both morphic stages. In this review, we take stock of the genes and metabolites reported for both morphic forms of *B. cinerea* between January 2015 and October 2022.

## 1. Introduction

The *Botrytis* genus includes a wide variety of pathogenic fungal species found throughout the world [1,2]. *Botrytis* species are characterised by grey mycelium and saprotrophic behaviour. They are necrotrophic pathogens responsible for heavy losses among many economically important horticultural and floral crops. Most *Botrytis* species are specialised pathogens with a narrow host range, generally infecting only one or a few closely related species within a single plant genus [3]. However, *Botrytis cinerea* is a polyphagous species that infects a wide range of host plants. 

Micheli in 1792 characterised the species *B. cinerea* for the first time, which was later confirmed by Persoon in 1801 [4,5,6]. Since then, a large number of species have been described thanks to progress in molecular genetics and the development of the relevant phylogenetic markers. Today, approximately 30 different *Botrytis* species have been identified. Seven new species, a hybrid, and a species complex have been isolated and identified over the last decade [7].

*B. cinerea* is the agent that causes grey mould disease which affects a total of 586 genera of vascular plants, representing over 1400 ornamental and agriculturally important plant species [8,9]. This fungus is a typical necrotroph whose infective cycle includes the killing of plant cells, maceration of plant tissue and then reproduction by forming asexual spores on the decomposing plant material.

The life cycle of *B. cinerea* can be divided in two phases generally described as asexual (anamorph) and sexual (teleomorph). During the asexual stage, which is defined vegetative, *B. cinerea* produces macroconidia, sclerotia and microconidia (spermatia). This is essentially the anamorph stage of *Botrytis* (*Botrytis cinerea* Pers). The sexual stage plays a key role in the sexual phase, indeed microconidia fertilise sclerotia to produce apothecia, in which meiosis occurs and ascospores are produced (the *Botryotinia* teleomorph stage named by Whetzel, 1945, *Botryotinia fuckeliana* (de Bary) Wetzel).

In a recent paper [10], Wingfield et al., tried to simplify fungi taxonomy, starting from the new molecular tools that make the split in morphs unnecessary. During the XVI International Symposium dedicated to the *Botrytis* fungus, which was held in Bari in June 2013, the scientific community unanimously decided to assign the most widely used name of the asexual form, *Botrytis*, to this genus of fungi [6]. The term “teleomorph” should, consequently, not be used. This taxonomic change was proposed by Johnston et al. and they provided *Botrytis* names for the only two *Botryotinia* species lacking a *Botrytis* equivalent [6,11].

Nevertheless, in the literature, we continue to find articles that describe the isolated metabolites of the teleomorph stage of *B. cinerea*. Recently, a short but interesting number of articles describing new metabolites isolated from *B. fuckeliana* have been published.

This study entailed a bibliographic search of databases such as PubMed, Scopus, Science Direct Elsevier, Google Scholar and especially the CAS SciFinder^n^ platform to retrieve information using keywords such as: “*Botrytis cinerea*”, “*Botryotinia fuckeliana*”, “secondary metabolites”, “sesquiterpene”, “diterpenes” and “polyketides” to find all the secondary metabolites isolated and reported from both morphic forms of *B. cinerea*. We also included the keywords “biosynthesis” and “biological activity” in our search criteria to locate information about the biosynthesis of the different families of metabolites described. In our search results, compounds which had been isolated and reported in the CAS SciFinder^n^ platform were indexed in this study and articles referencing that type of compound were analysed. In order to exclude some of the articles, automatic search tools were used, and others were screened manually. Papers published in Chinese and Japanese languages were excluded from the analysis, except when an extensive summary of the article was available in English.

This review provides an overview of publication trends on the genes and metabolite structures described for both morphic forms of *B. cinerea* between January 2015 and October 2022. A total of 100 references were analysed.

## 2. *Botrytis cinerea* Pers. Fr. (The Anamorph Stage)

Undoubtedly, the most widespread species of the genus *Botrytis* is *B. cinerea* Pers. Fr. (Sclerotiniaceae), considered the most important postharvest fungus affecting a large number of commercial crops, including some of the most economically important ones in Andalusia, Spain [1]. This fungus is ranked second among the top ten phytopathogenic fungi in terms of scientific and economic importance [12].

This fungus infects host cells by producing toxins and reactive oxygen species and triggering oxidative bursts [13]. An interesting review of the literature was performed up to December 2014 on secondary metabolism in *B. cinerea*, combining genomic and metabolomic approaches [14].

Two families of nonspecific phytotoxins are produced by *B. cinerea* [14]: botryane-type sesquiterpenes, mainly botrydial (**1**) and dihydrobotrydial (**2**), their relatives [14,15], and polyketides such as botcinic and botcineric acids (**3**,**4**) and their botcinin cyclic derivatives (**5**,**6**) [14,16,17] (Figure 1). Both groups of compounds induce chlorosis and cell collapse which appears to facilitate the penetration and colonization of plant tissue [15,18,19]. Botrydial (**1**) has been considered a pathogenicity factor of the fungus, as it has also been detected during plant infection [18].

Moreover, the sesquiterpene abscisic acid (ABA) (**7**), which is involved in leaf fall in plants [20], and a few structurally related derivatives [14] are also produced by B. cinerea. The polyketides botrylactone (**8**) [21] and cinbotolides A and B (**9**-**10**) [22] have been described. Furthermore, the peptide siderophore ferrirhodin (**11**) [23], metabolites with a mixed biosynthetic origin such as 4β-acetoxytetrahydrobotryslactone (**12**) [24] and the plant growth inhibitor cinereain (**13**) [25] have been reported. Some other interesting orphan metabolites isolated from *Botrytis* spp. were gleaned from a review by Collado and Viaud, 2016 [14].

### 2.1. From Genes to Molecules

Although the identification of the genes involved in the biosynthetic pathways to botrydial (**1**) and abscisic acid (ABA) (**7**) started already two decades ago [26,27,28], the sequencing of the *B. cinerea* genome provided a complete picture of all genes involved in the biosynthesis of secondary metabolites (SMs) [29,30] and aided research on the genetic determinants of SM production.

The recent resequencing of the genome of *B. cinerea*, strain B05.10, and its update in the Emsembl Fungi database (https://fungi.ensembl.org/Botrytis_cinerea/Info/Index (accessed on 1 December 2022)) [31,32] revealed a total of 44 genes encoding key enzymes involved in the biosynthetic routes of secondary metabolites of the phytopathogen, including STCs (sesquiterpene cyclases), DTCs (diterpene cyclases), PKSs (polyketide synthases), NRPSs (non-ribosomal peptide synthetases), PKS-NRPS hybrids and DiMethylAllylTRyptophan Synthases (DMATS). These genes were predicted from the sequences of the B05.10 and T4 strains. 

Genomic data has revealed that the plant pathogenic fungus *B. cinerea* appears to have seven STC (*Bcstc*) genes that encode proteins with the typical two magnesium binding sites required for farnesyl diphosphate (FDP) (**14**) cyclisation. Four STCs have been functionally characterised to date: *Bcstc1* [33], *Bcstc5* [34], *Bcstc7* [32] and the rare new class of STC as an Open Reading Frame (ORF), the *Bcaba3* gene, involved in abscisic acid (ABA) biosynthesis [35].

#### 2.1.1. Botrydial Gene Cluster

As already indicated, Botrydial (**1**) is a phytotoxic metabolite [13,14] which reproduces the symptoms of the *Botrytis* infection [15]. This compound is considered to be one of the main toxins of *B. cinerea* [13,18] and it was one of the first metabolites to be isolated from this fungus. 

The botrydial biosynthetic gene cluster has been identified (Figure 2) [27,33,36]. The cluster consists of seven genes (*Bcbot1* to *Bcbot7*) (for *Botrytis cinerea* botrydial biosynthesis) coding for a sesquiterpene cyclase (*Bcbot2*), an acetyltransferase (*Bcbot5*) and three monooxygenases (*Bcbot1*, *Bcbot3* and *Bcbot4*), typical of secondary metabolism in filamentous fungi. Subsequently, a transcription factor, Zn(II)_2_Cys_6,_ (*Bcbot6*), and a gene encoding a dehydrogenase (*Bcbot 7*) were reported [37].

The *Bcstc1* gene, also named *Bcbot2*, is responsible for the corresponding cyclisation step of FDP (**14**) in the biosynthetic route to botrydial (**1**). Targeted deletion of the *Bcbot2* gene therefore stopped production of botrydial (**1**) and all related probotryane metabolites. Direct evidence for the biochemical function of *Bcbot2* came from the demonstration that recombinant *BcBot2* protein converted farnesyl diphosphate (FDP) (**14**) to the parent tricyclic alcohol of presilphiperfolan-8-ol (PSP) or probotryan-9β-ol. The structures and numbering systems for the presilphiperfolanes and botryanes were established before their biosynthetic relationships were known. 

The process continues with the involvement of an acyltransferase and three cytochrome P450 monooxygenases encoded by genes *Bcbot 1, 3* and *4* [27,36]. The high degree of homology observed between the amino acid sequences obtained from *Bcbot5* and acetyltransferases suggests that this gene encodes the enzyme responsible for the introduction of the acetyl group in probotryanes in the final stages of biosynthesis [36]. The gene *Bcbot6* encodes the transcription factor Zn(II)_2_Cys_6_, which is responsible for the regulation of the entire toxin gene cluster, while *Bcbot7* produces a dehydrogenase that may be involved in the conversion of botrydial (**1**) into dihydrobotrydial (**2**) [37] (Figure 2). The identification of *Bcbot6* as the main regulator of botryane synthesis is the first step towards a more comprehensive understanding of the whole regulation network of botrydial (**1**) and relative biosynthesis of its ecological role in the *B. cinerea* life cycle [37].

#### 2.1.2. ABA Gene Cluster

ABA (**7**) is a plant hormone that plays an important role in many aspects of plant growth and development and in the initiation of adaptive responses to various environmental conditions [38,39]. ABA (**7**) is mainly produced by plants but is also produced as a secondary metabolite by several species of filamentous fungi such as the genera *Botrytis*, *Penicillium*, *Cercospora* and *Rhizoctonia* [40]. Studies have shown that the ABA biosynthetic pathway differs between plants and fungi [41]. The understanding of the molecular mechanism driving ABA biosynthesis in fungi remains limited, especially for the steps from FDP (**14**) to ABA (**7**) [42].

In 2006, the biosynthetic gene cluster (BcABA), consisting of four putative enzyme genes (*Bcaba1*–*Bcaba4*), was identified in *B. cinerea*. *Bcaba1* and *Bcaba2* are cytochrome P450s. *Bcaba3* shows no homology to functionally characterised enzymes while *Bcaba4* is similar to a short-chain dehydrogenase/reductase (Figure 3A) [28]. Knock-out experiments confirmed that *Bcaba1, 2* and *3* are essential for ABA production in *B. cinerea*, whereas *Bcaba4* is involved in the pathway but is not essential [26,28]. It was hypothesised that the cyclisation of FDP (**14**) requires a sesquiterpene cyclase (STC); however, none of the proteins in the gene cluster exhibited known STC motifs. A study by Izquierdo-Bueno et al. identified an STC gene called *Bcaba5*, which is co-expressed with, but not located in, the gene cluster [34].

Five STC-coding genes were identified thanks to the complete genome sequencing of the ABA-producing strain ATCC58025. Among them, *Bcstc5* exhibits an expression profile coinciding with ABA production. Gene inactivation, complementation and chemical analysis demonstrated that *BcStc5/BcAba5* is the key enzyme responsible for the key step of ABA biosynthesis in fungi, located on chromosome 1 in *B. cinerea* strain B05.10 (Figure 3B). This gene is involved in the cyclisation of FDP (**14**) into 2*Z*,4*E*-α-ionylidene-ethane (Figure 4). Hence, an ABA cluster formed by five genes is proposed [34].

Nevertheless, Takino et. al. [35] suggested that the *Bcaba3* gene participates in skeletal construction for the formation of ABA (**7**), identifying this gene as an α-ionylideneethane synthase and thus revealed a three-step reaction mechanism through a series of biotransformation experiments, heterologous expression experiments, and in vitro enzymatic studies. They also showed that ABA (**7**) could be produced heterologously in an *Aspergillus oryzae* strain expressing *Bcaba1, 2, 3* and *4* [35]. On the other hand, Otto et. al. [43] constructed and characterised an ABA-producing *S. cerevisiae* strain using the ABA biosynthetic pathway from *B. cinerea*, expression of the *B. cinerea* genes *Bcaba1, 2, 3* and *4* being sufficient to establish ABA production in the heterologous host [43]. The contradicting results of these two studies raise the question as to whether *Bcaba3* and *Bcaba5* catalyse the same reaction. Consequently, the gene involved in the cyclisation of FDP (**14**) for the formation of ABA (**7**) is still unknown. 

In contrast, the most recent studies on ABA (**7**) have shown that the putative methyltransferase LaeA/LAE1 plays an important role in the regulation of ABA biosynthesis in *B. cinerea*. The deletion of *Bclae1* caused a 95% reduction in ABA yields, accompanied by a decrease in the transcriptional level of the ABA synthesis gene cluster *Bcaba1-4*. Further RNA-seq analysis indicated that the deletion of *Bclae1* also affected the expression level of key enzymes of botcinic acid (BOA) and botryanes (BOT) in secondary metabolism and accompanied clustering regulatory features showing that this gene is a global regulator involved in the biosynthesis of a variety of secondary metabolites in filamentous fungi [44].

#### 2.1.3. Eremophilenol Genes

A new family of cryptic metabolites with a (+)-4-*epi*-eremophil-9-en-11-ol skeleton was biosynthesised by *B. cinerea* when sublethal doses of CuSO_4_ (5 ppm) were added to the culture media [45,46]. Studies on the in vitro evaluation of the biological role of these metabolites showed their involvement in the self-regulation of asexual spore production and enhanced the production of complex appressoria (infection cushions). This fact indicated for the first time the participation of sesquiterpenoid metabolites in the regulation of infective structures. Moreover, these metabolites possess an enantiomeric carbon skeleton resembling that of phytoalexin capsidiol suggesting that eremophilenols may be effectors that inhibit plant defences or modulate plant immunity to enhance the infection process [46].

PacBio technology, and the resulting update of the Ensembl Fungi (2017) database in the genome sequence, was instrumental in the identification of new possible genes that could be involved in secondary metabolism. As a result, a new STC coding gene (*Bcstc7*) has been included in the gene list from this plant pathogenic fungus (Table 1). An expression study of *B. cinerea* genes that encode sesquiterpene cyclases (*Bcstc1−5* and *Bcstc7*) showed that gene *Bcstc7* produced a more important level of expression when compared with the other STC-encoding genes. Metabolomic characterisation showed that *ΔBcstc7* was impaired in the production of eremophilenol derivatives while the complemented transformant *^compl^ΔBcstc7^niaD^* recovered its capacity to produce this family of compounds, demonstrating that this gene is the key enzyme responsible for the cyclisation of FDP (**14**) to eremophil-9-en-11-ols (Figure 4) [32].

The sesquiterpene cyclases STC2, STC3, STC4 and STC6 of the *B. cinerea* fungus remain uncharacterised to date. Nevertheless, gene *Bcstc6* (ID P020710.1) is specific to the T4 strain and absent in many of the other strains studied, including B05.10 [29].

#### 2.1.4. Retinal and Carotenoid Gene Cluster

Diterpene cyclases catalyse the cyclisation of the linear 20-carbon substrate geranylgeranyl diphosphate to produce the diterpene scaffold occurring via a carbocation cascade. This ionisation-dependent reaction is catalysed by class I terpene synthases [47,48]. 

The diterpene cyclases DTC1, DTC2, DTC3 and PAX1 of the *B. cinerea* fungus remain uncharacterised to date. However, *B. cinerea* also contains a Terpene Synthase (TS) encoding gene (PAX1), orthologous to the *Penicillium paxilli PaxC* gene, involved in the biosynthesis of the indole-diterpene paxillin [29,49]. It is therefore very likely that this gene is involved in the production of a yet to be discovered indole-terpene for this phytopathogen.

Studies carried out by Schumacher et al. [50] on the transcription factor BcLTF1 have shown that its absence affects the expression of secondary metabolism-related genes. The expression patterns of only three key enzyme-encoding genes (*Bcnrps2*, *Bcphs1*, *Bcstc5*) were significantly affected by light. Notably, one of them corresponds to a cluster of light-induced genes (*Bcphs1*, *Bcphd1*, *Bccao1*) that encode homologues of enzymes involved in the biosynthesis of retinal, the chromophore for opsin, in *Fusarium fujikuroi* (phytoene synthase, phytoene dehydrogenase, and carotenoid oxygenase). In that publication, the authors proposed a pathway for carotenoid biosynthesis and *B. cinerea* gene clusters for retinal biosynthesis organised in the same way as for *F. fujikuroi* (Figure 5) [50]. There are no reports describing the isolation of carotenoids from *B. cinerea*. However, β-carotene was isolated from *Sclerotinia* spp. [51]. This research suggested that the *Bcphs1* gene is related to the production of retinal diterpene in *B. cinerea*.

#### 2.1.5. Botcinic Acid and Botcinins Gene Cluster

PKSs found in fungi, bacteria and plants are large megasynthases related to fatty acid synthases that biosynthesise small molecule polyketides with diverse natural functions and include well known secondary metabolites [52]. 

The iterative nature of fungal PKSs means that, in most cases, there is only one PKS involved in the synthesis of a particular polyketide. However, some fungal polyketides are known to be assembled by the action of two PKSs [53].

This is the case of botcinic acid and its relatives, botcinins, which require the action of two PKSs, *BcBoa6* and *BcBoa9* [53]. The expression study of the 20 *BcPKS* genes predicted from the genome sequence of strain B05.10 was studied at different physiological stages. During infection, only *Bcpks6* and *Bcpks9* exhibited higher levels of expression than the actin gene. The study of *ΔBcboa6* and *ΔBcboa9* null mutants did not produce botcinic acid or its derivatives, indicating that they act in concert to synthetise botcinic acid. Additionally, these authors proposed a cluster for *B. cinerea* botcinic acid biosynthesis made up of 17 putative biosynthetic genes (*Bcboa1* to *Bcboa17*) (Figure 6) [53]. Later studies showed that *BcBoa13*, a putative Zn_2_Cys_6_ transcription factor, is a nuclear protein playing a major positive regulatory role in the expression of other *Bcboa1*-to-*Bcboa12* genes and botcinic acid production [54].

#### 2.1.6. Melanin Gene Cluster

It has been reported that the accumulation of 1,8-dihydroxynaphthalene (DHN) melanin is responsible for the pigmentation of the macroconidia and/or black sclerotia in *B. cinerea*. Schumacher carried out an interesting study which described the genetic basis and regulation of DHN melanogenesis in *B. cinerea* [55]. This author identified and functionally characterised the putative melanogenic and regulatory genes. Unlike other DHN melanin-producing fungi, *B. cinerea* and other Leotiomycetes contain two key (PKS)-encoding enzymes. *Bcpks12* and *bcpks13* are developmentally regulated and are required for melanogenesis in sclerotia and conidia, respectively (Figure 7). Regulation of the melanogenic genes involves three pathway-specific transcription factors (TFs), BcSMR1, BcZTF1 and BcZTF2 (Figure 7). These are clustered with *bcpks12* or *bcpks13* and other developmental regulators such as light-responsive TFs. Melanogenic genes are dispensable in vegetative mycelia for proper growth and virulence [55,56]. 

Recently, an exhaustive review about insights from genes studied with mutant analysis in *B. cinerea* has been published. The biosynthesis of DHN melanin pathways in *B. cinerea* has been reviewed [57]. 

Deletion of the *Bcpks12* and *Bcpks13* genes resulted in albino sclerotia and conidia in *B. cinerea*, indicating complete melanogenesis disruption [55,56] and studies so far indicate that melanisation does not significantly affect pathogenicity or fungal development [57].

#### 2.1.7. Pyrones, Resorcylic Acids and Resorcinols

Another gene of this family that has been studied is the *Bcchs1/Bcpks* gene, which was characterised by Jeya et al. [58]. BPKS from *B. cinerea* is a novel type III polyketide synthase that accepts C4–C18 aliphatic acyl-CoAs and benzoyl- CoA as starters to form pyrones, resorcylic acids and resorcinols by sequential condensation with malonyl-CoA (Figure 8). This PKS shows the highest catalytic efficiency ever reported for a long chain acyl-CoA ester [58].

The functionality of 14 PKSs in *B. cinerea* is still unknown, the only thing certain being that they will code for polyketides that are also unknown.

#### 2.1.8. Siderophore Genes

Non-ribosomal peptide synthetases (NRPSs) are multi-modular enzymes, found in fungi and bacteria, which biosynthesise peptides without the aid of ribosomes. Bushley and Turgeon [59] identified genes (NPS) encoding NRPS and NRPS-like proteins in 38 fungal genomes and undertook phylogenomic analyses in order to identify fungal NRPS subfamilies, assess taxonomic distribution, evaluate conservation levels across subfamilies and address evolutionary mechanisms of multi-modular NRPSs. Phylogenomic analysis identified major subfamilies of fungal NRPSs which fall into two main groups: 1) a group of primarily mono/bi-modular enzymes containing the PKS-NRPSs and 2) a group of primarily multi-modular proteins, siderophore synthetases (SID) and Euascomycete-only synthetases (EAS) which appear both restricted to and highly expanded within fungi [59].

All fungal PKS-NRPS hybrids fall into a single, well supported, monophyletic group, which suggests a single origin. In *B. cinerea,* three PKS-NRPS hybrids, five EAS and four SID were identified. Of these latter enzymes, EAS and SID belong to the NRPS family. NRPS 2,3,7 are possibly involved in the production of secondary metabolites such as ferrichrome siderophores and NRPS 6 coprogene siderophore [59]. However, the production of secondary metabolites of 5 NRPS, and the 2 DMATS of *B. cinerea,* is still unknown. 

The key enzymes related to secondary metabolism in *B. cinerea* identified to date are summarised in Table 1.
plants-12-00553-t001_Table 1Table 1Repertoire of secondary metabolism key enzymes (KEs) in *B. cinerea* (B05.10).Gene Name(s)Gene ID in NCBIGene ID in EnsemblFungiMetabolite When Isolated or PredictedSTCs (sesquiterpene cyclases)*Bcstc1/Bcbot2*BC1G_16381Bcin12g06390Botrydial [33]*Bcstc2*BC1G_09560Bcin08g02350Unknown sesquiterpenes*Bcstc3*BC1G_06357Bcin13g05830*Bcstc4*BC1G_14308Bcin04g03550*Bcstc5*BC1G_10537Bcin01g03520Abscisic acid [34]*Bcstc7*BC1G_12849Bcin11g06510Eremophil-9-en-11-ols [32]DTCs (diterpene cyclases)*Bcdtc1*BC1G_13295Bcin01g04920Unknown diterpenes*Bcdtc2*BC1G_06148
*Bcdtc3*BC1G_06751Bcin08g03560*Bcpax1*BC1G_01823Bcin05g05670Unknown indole-diterpene*Bcphs1*BC1G_13908Bcin01g04560Retinal [50]PKSs (polyketide synthases)*Bcpks6/Bcboa6*BC1G_16087Bcin01g00060Botcinic acid [53,60]*Bcpks9/Bcboa9*BC1G_15839Bcin01g00090*Bcpks12*BC1G_06876Bcin02g08770Melanin [55]*Bcpks13*BC1G_14497Bcin03g08050*Bcpks1*BC1G_13366Bcin14g00600Unknown polyketides*Bcpks2*BC1G_02586Bcin02g01680*Bcpks4*BC1G_04786Bcin11g02700*Bcpks8*BC1G_02704Bcin07g02920*Bcpks10*BC1G_04310Bcin13g01510*Bcpks11*BC1G_09042Bcin14g01290*Bcpks14*BC1G_08227Bcin16g01830*Bcpks15*BC1G_01752Bcin05g06220*Bcpks16*BC1G_10687Bcin16g05040*Bcpks17*BC1G_01953Bcin03g02010*Bcpks18*BC1G_06884Bcin02g08830*Bcpks19*BC1G_16074Bcin08g00290*Bcpks20*BC1G_07030Bcin04g00640*Bcpks21*BC1G_13114Bcin05g08400*Bcchs1/Bcpks*BC1G_06032Bcin13g02130Pyrones, resorcylic acids andresorcinols [58]**Gene Name(s)****Gene ID in NCBI****Gene ID in EnsemblFungi****Metabolite When Isolated or Predicted**PKS-NRPS hybrids*Bcpks3*BC1G_00695Bcin03g04360Unknown amino-acid containing-polyketides (PKS-NRPS hybrids)*Bcpks5*BC1G_15479Bcin01g11550*Bcpks7*BC1G_15702Bcin10g00040NRPSs (non-ribosomal peptide synthetases)*Bcnrps2*BC1G_03511Bcin12g00690Ferrichrome siderophores [59]*Bcnrps3*BC1G_10928Bcin16g03570*Bcnrps7*BC1G_15494Bcin01g11450*Bcnrps6*BC1G_10567Bcin01g03730Coprogene siderophore [59]*Bcnrps1*BC1G_07441Bcin12g04980Unknown peptides*Bcnrps4*BC1G_02495Bcin02g02380*Bcnrps5*BC1G_10622Bcin04g01390*Bcnrps8*BC1G_04782Bcin11g02650*Bcnrps9*BC1G_09041Bcin14g01300DMATS (DiMethylAllylTryptophan Synthases)*Bcdmats1*BC1G_08209Bcin16g01940Unknown alkaloids*Bcdmats2*BC1G_15920Bcin14g04900


### 2.2. New Insights in the Secondary Metabolism of B. cinerea, January 2015 to October 2022

#### 2.2.1. Botrydial Biosynthetic Gene Cluster Studies

As indicated in Section 2.1.1 of this review, botrydial (**1**) was the first metabolite isolated from *B. cinerea.* The characterisation of **1** and dihidrobotrydial (**2**) led to the identification of many other derivative compounds [61] previously described and reviewed by our research group [14]. The biosynthesis pathway of this family of compounds has been established [62,63,64,65]*,* and more recently, the mechanism and stereochemistry of the enzymatic formation of the probotryane precursor, presilphiperfolan-8β–ol (PSP), was reported by Cane et al. [66].

The botrydial biosynthetic gene cluster has been identified (Figure 2) [27,33,37]. This cluster consists of seven genes (*Bcbot1* to *Bcbot7*) coding for a sesquiterpene cyclase (*Bcbot2*), an acetyltransferase (*Bcbot5*), a transcription factor (*Bcbot6*), a putative dehydrogenase (*Bcbot7*) and three monooxygenases (*bcbot1*, *bcbot3* and *bcbot4*) [33,36,37]. *Bcbot1* was identified as the first gene, coding for a P-450 monooxygenase, that was involved in botrydial synthesis [27] and catalysing hydroxylation at C-15 of the probotryane intermediate, PSP (Figure 9).

Recently, the genetic and molecular basis of botrydial biosynthesis have been described [36,37]. Genes *Bcbot3* and *Bcbot4* were deleted by homologous recombination and showed to catalyse regio- and stereospecific hydroxylation at the C-10 and C-4 carbons, respectively, of the probotryane intermediate skeleton (Figure 9) [36].

A detailed study of the *∆bcbot4* null mutant was undertaken in order to discover the metabolic fate of the PSP intermediate biosynthesised by *B. cinerea* after longer periods of fermentation [67]. This led to the identification of three new presilphiperfolanes (**15**-**17**) and three new cameroonanes (**18**-**20**) (Figure 10 (A)). The rearrangement to cameroonanes was facilitated by the absence of hydroxylation at C-11, whereas functionalisation at this position precludes this rearrangement. This could suggest that the interactions of the C-11 hydroxylated derivatives hinder the stereo-electronic requirements for the migration of the C-11:C-7 sigma bond to C-8 (Figure 10 (B)) [67].

Another study examined the metabolism of botryane sesquiterpenoids of *B. cinerea* [68]. The study of metabolites with botryane and presilphiperfolane skeleton of the fungus *B. cinerea* has shed light on the biosynthesis of this family of sesquiterpenoids and has also led to potentially novel approaches to the selective control of this pathogen. An ecological role of the naturally occurring sesquiterpenoids in terms of their effect on the growth of these plant pathogens has been suggested [68].

#### 2.2.2. Botrydial Applications

A phospholipid second messenger called phosphatidic acid (PA) is involved in the stimulation of plant defence mechanisms. It is produced by either phospholipase D (PLD) or by the concurrent activity of phospholipase C and diacylglycerol kinase (PLC/DGK), two different enzyme mechanisms. Through PLD and PLC/DGK, botrydial (**1**) causes the production of PA in a matter of minutes [69]. Both PLC and DGK inhibition reduce ROS production sparked by botrydial (**1**). This shows that PLC/DGK is upstream of ROS production. PLC is encoded in tomato by the multigene family SlPLC1–SlPLC6 and the pseudogene SlPLC7. It has been shown that plants that lack SlPLC2 are less vulnerable to *B. cinerea*. Additionally, by specifically engineering a microRNA to silence the expression of SlPLC2, it has been possible to investigate the impact of SlPLC2 on botrydial-induced PA generation. SlPLC2-silenced-cell suspensions generate PA levels comparable to those of wild-type cells after botrydial treatments. It is safe to say that PA is a novel element resulting from the responses that botrydial (**1**) causes in plants [69].

The function of botrydial (**1**) in the interaction between the phytopathogenic fungus *B. cinerea* and bacteria associated with plants was examined. Nine types of bacteria found in soil and phyllospheric samples were shown to be susceptible to growth-inhibition caused by *B. cinerea*. The lack of bacterial inhibition induced by *B. cinerea* mutants incapable of producing botrydial (**1**) demonstrated the inhibitory function of botrydial (**1**). Via taxonomic research, these bacteria were identified as belonging to several *Bacillus* species (six strains), *Pseudomonas yamanorum* (two strains) and *Erwinia aphidicola* (one strain). *Bacillus amyloliquefaciens* strain MEP_2_18 and WT, and *B. cinerea* mutants that do not produce botrydial, were inoculated together in soil to show that both microbes exert reciprocal inhibitory effects, *B. cinerea*’s inhibition was dependent on botrydial production. Furthermore, the presence of *B. amyloliquefaciens* MEP_2_18 in in vitro confrontation assays the affected formation of botrydial (**1**). In turn, purified botrydial (**1**) prevented *B. amyloliquefaciens* MEP_2_18 from producing cyclic lipopeptide (surfactin) and *Bacillus* strains from developing in vitro. Overall, findings show that *B. cinerea* has the capacity to suppress potential biocontrol *Bacillus* genus bacteria due to botrydial (**1**). It has been suggested to include resistance to botrydial (**1**) among the criteria determining the choice of biocontrol agents for plant diseases brought on by *B. cinerea* [70].

#### 2.2.3. New Polyketides from B. cinerea

As described in references [36] and [71], the *bcbot4∆* mutant also overproduced a significant number of polyketides which included, in addition to known botcinins, botrylactones and cinbotolide A (**9**), two new botrylactones (**21**,**22**) and two cinbotolides, cinbotolides B (**10**) and C (**23**) (Figure 11) [36]. A subsequent detailed study of the polyketides produced by the null mutant *bcbot4∆* [71] led to the characterisation of five new polyketides, three derived from botcinic and botcineric acids: botcinins H (**24**), I (**25**), J (**26**), one derived from the initial pentaketide: botcinin K (**27**) and one cinbotolide derivative: cinbotolide D (**28**) [71]. The structural characterisation of botcinin K (**27**) [71] showed a basic chemical structure corresponding to a botcinin (C14) derivative obtained directly from the original per-methylated pentaketide leading to the biosynthesis of botrylactone and other botcinins, confirming the previously proposed biosynthetic route [60,71].

The cluster of *Bcboa* genes responsible for botcinins biosynthesis was found to be specifically regulated [54], and it was discovered that this cluster is situated in a subtelomeric genomic region. According to genetic studies, *BcBoa13*, a putative Zn_2_Cys_6_ transcription factor, is a nuclear protein that plays a major positive regulatory role in the expression of *Bcboa1* through *Bcboa12* genes and botcinic acid production [54]. Interestingly, the structure and regulation of the botcinic acid gene cluster share characteristics with the cluster responsible for the biosynthesis of the other known phytotoxin produced by *B. cinerea*, the sesquiterpene botrydial (**1**) [54]. This study demonstrated that the Zn_2_Cys_6_ protein Bcboa13 positively controls the clustered *Bcboa1-Bcboa12* genes y consequently the synthesis of botcinic acids and other botcinins in *B. cinerea* [54]. 

#### 2.2.4. Abscisic Acid Biosynthetic Gene Cluster Studies

Abscisic Acid (ABA) (**7**) is a well-known hormone produced by plants through the carotenoid’s pathway. Surprisingly, this sesquiterpene can also be produced by a small number of fungi including the plant pathogenic species *B. cinerea* [20]. However, the ABA biosynthetic pathway in fungi differs from the carotenoid pathway described in plants. Hence, in *B. cinerea*, it has been demonstrated that ABA (**7**) is obtained from the cyclisation of farnesyl diphosphate (FDP) (**14**) and subsequent oxidation steps. Inomata et al. therefore proposed a biosynthetic pathway that involves the transformation of FDP (**14**) to give 2*Z*,4*E*,6*E*-allofarnesene which is then cyclised to form 2*Z*,4*E*-α-ionylideneethane [72]. This intermediate is then subjected to several oxidative steps to form ABA (**7**).

However, most *B. cinerea* species produce low or even undetectable levels of ABA (**7**) in vitro. The first gene involved in the biosynthesis of abscisic acid (**7**) was identified by Siewers et al., using an over-producer isolate of *B. cinerea* [26]. The *Bcaba1* gene was over-expressed in the presence of mevalonic acid in the medium. In addition, mutants deleted in the *Bcaba1* gene were impaired in terms of ABA production. The genomic locus includes three other genes (*Bcaba2-4*) that codify for a P450 monooxygenase, a putative dehydrogenase/reductase, and an unknown protein. However, the neighbouring genomic region does not have a gene that codifies for a sesquiterpene cyclase. Targeted inactivation of the genes proved the involvement of *Bcaba2* and *Bcaba3* in ABA biosynthesis and suggested a contribution of *Bcaba4* [28]. The close linkage of these four genes served as strong evidence for the presence of an abscisic acid gene cluster in *B. cinerea*.

Questions have arisen around why a plant hormone would be synthetised by a fungus and this is why a negative regulator of disease resistance through the down-regulation of defence response has been considered for ABA (**7**) [73]. The ABA (**7**) knock-out mutant proved to be more resistant to *B. cinerea*, swiftly fortifying the epidermal cell wall and exhibiting a higher induction of some genes involved with defence against *B. cinerea*. However, the knock-out mutant of the *Bcaba1* gene for *B. cinerea* showed that it is not necessary for fungal virulence [74].

As already mentioned, three abscisic acid biosynthetic studies have been conducted in recent years, but have produced conflicting results. First of all, to identify the STC responsible for the biosynthesis of ABA (**7**) in fungi, a genomic approach to *B. cinerea* was taken by Izquierdo-Bueno et al. [34]. Inactivation of the *Bcstc5*/*Bcaba5* gene in the *B. cinerea* ABA-overproducing strain ATCC58025 abolished ABA production. However, the complemented mutant restored the production of the sesquiterpene, demonstrating that the encoded STC was essential for ABA biosynthesis [34].

However, a more recent study by Takino et al. [35] showed that the four genes of the gene cluster, *Bcaba1*, *2*, *3* and *4*, are sufficient to produce ABA (**7**) in *Aspergillus oryzae* [35], thereby contradicting the finding of Izquierdo-Bueno et al. [34]. They also studied *Bcaba3*, a gene of the original gene cluster encoding an enzyme with hitherto unknown functions and no known motifs. In vitro assays showed that BcABA3 can convert FDP (**14**) to α-ionylideneethane [35]. Concerning the role of the *Bcaba5* gene in the ABA biosynthetic pathway, Takino et al., suggested that this gene could be involved in ABA biosynthesis but is not essential.

These studies were reinforced by a heterologous expression of the ABA biosynthetic cluster, *Bcaba1-4,* in *S. cerevisiae*, by Otto et al., highlighting the importance of the *Bcaba3* gene in the biosynthesis of this metabolite [43]. This work confirmed the finding of Takino et al., i.e., that the four genes in the *B. cinerea* ABA gene cluster *Bcaba1-4* are sufficient to produce ABA (**7**) and *Bcaba5* is not essential. However, it is still possible that the co-expression of *Bcaba5* enhances ABA production, as in the case of artemisinic acid biosynthesis [75]. Indeed, *Bcaba5* and the monooxygenase *BcceP450* are expressed during ABA production in *B. cinerea* indicating that they could be involved in the pathway [34]. Nonetheless, further analysis is needed to definitively confirm that BcABA5 and monooxygenase BcceP450 are not involved in ABA biosynthesis [34,35,43].

Interestingly, Takino et al. have discovered a brand-new class of sesquiterpene synthase as an Open Reading Frame (ORF) (BCIN_08g03880) called *Bcaba3* and its unusual three-step reaction process involving two neutral intermediates called β-farnesene and allofarnesene. Database searches revealed that the homologous enzyme genes are present in more than 100 bacteria and that *BcABA3* is not homologous with normal sesquiterpene synthases, indicating that these enzymes belong to a new family of sesquiterpene synthases [35].

Lastly, the function of the four biosynthetic genes *Bcaba1-Bcaba4*, found in *B. cinerea* through biotransformation experiments and in vitro enzymatic reactions, was elucidated.

A biosynthetic pathway was postulated and represented in Figure 12 [76].

The first interesting step is the cyclisation of FDP (**14**) to give α-ionylideneethane (**29**) catalysed by a novel sesquiterpene synthase, BcABA3, which exhibits low amino acid sequence identities with sesquiterpene synthases. Subsequently, two cytochrome P450s, BcABA1 and BcABA2, catalysed the first two reactions, mediating oxidative modifications of the cyclised product to afford 1ʹ,4ʹ-*trans*-dihydroxy-α-ionylideneacetic acid (**32**), which undergoes alcohol oxidation to furnish ABA (**7**).

One of them, BcABA2, is notable because it catalyses two rounds of allylic oxidation, the first oxidising the β-face of C4’ and the second oxidising the α-face of C1´. The intermediate **30** also undergoes allylic oxidations which are also mediated by enzymes found naturally in *A. oryzae*. However, when the downstream oxidation enzyme genes, *Bcaba2* and *Bcaba4*, were co-expressed with *Bcaba1*, this unexpected oxidation was unable to impair the production of **7**, as the swift conversion of **30** into **32** stifled the side reaction. The fact that BcABA4 can accommodate diastereomeric molecules **32** and **34**, even though **32** converts at a significantly higher rate than **34**, makes the alcohol oxidation of **32** catalysed by BcABA4 fascinating [76].

#### 2.2.5. Eremophilenes Isolated from B. cinerea

The high number of key secondary metabolism enzymes encoded in the genome of *B. cinerea* does not correspond to the total number of chemically characterised metabolites for this fungus. Thus, *B. cinerea* has a high number of silenced gene clusters (not expressed) under standard laboratory conditions. 

The expression of silent biosynthetic pathways could be induced by means of a chemically based epigenetic approach, OSMAC (One Strain—Many Compounds) methodology, or by a molecular approach through gene inactivation. Following an OSMAC approach, by chemical induction with copper sulphate, a cryptic sesquiterpenoid family with (+)-4-*epi*-eremophil-9-en-11-ol (**35**-**47**) [32,45,46], 11-hydroxyeremophil-1(10)-en-2-one (**48**-**50**) [32] and 11,12,13-tri-*nor*-eremophilene (**51**-**56**) [77] skeletons has been reported (Figure 13).

The study of the biological role of these metabolites led to the conclusion that eremophilenols (**35**-**41**) were involved in the autoregulation of asexual spore production and enhanced the production of complex appressoria (infection cushions) [45]. This indicated for the first time that sesquiterpenoid metabolites participated in the regulation of infective structures. Furthermore, they may be potential effectors used by *B. cinerea* to circumvent plant chemical defences against plant pathogenic fungi [46].

The biosynthetic route of these eremophilenes from (*E*,*E*)-FDP (**14**), using deuterium and carbon-13 labelled acetate, has been reported [46] (Figure 14). Additionally, Suárez et al. [32] have recently identified the sesquiterpene cyclase STC7 (see Section 2.1.3) involved in the cyclisation of (*E,E*)-FDP (**14**) to the germacrene intermediate (*S*)-hedycaryol. Cyclisation of its DU conformer yielded (+)-4-*epi*eremophil-9-en-11-ol derivatives via a *cis*-fused eudesmane cation (Figure 14).

The null and complement transformants in STC7 were recently constructed, enabling the functional characterisation of this STC [32,77]. Deletion of the Bcstc7 gene abolished (+)-4-*epi*-eremophilenol biosynthesis, which could then be re-established by complementing the null mutant with the Bcstc7 gene [32].

A thorough analysis of the metabolites produced by two wild-type strains, B05.10 and UCA992, and the complemented mutant ^compl^∆Bcstc7^niaD^ revealed the isolation and structural characterisation of six 11,12,13-tri-*nor*-eremophilene derivatives (**51-56**), in addition to a high number of known eremophilen-11-ol derivatives [77]. A structural characterisation was carried out by means of extensive spectroscopic techniques. The biosynthesis of these compounds has been reported (Figure 15) and explained by a retroaldol reaction of eremophilenol (**A**) to give the keto derivative **B** which, after oxidation at C-7, would yield the corresponding 8-keto-7-hydroxy derivative C, corresponding to compounds **51-53**; or by dehydration and oxidative cleavage of C11-C13 carbons which, after reduction, would yield F compounds **54-56** [77]. The structures, occurrences and biosynthesis of 11,12,13-tri-*nor*-sesquiterpenes have recently been reviewed [78].

#### 2.2.6. Other Metabolites Isolated from B. cinerea

A new cryptic metabolite, botrycinereic acid (**57**) (Figure 16) was isolated by chemical epigenetic manipulation of *B. cinerea* strain B05.10 with the histone deacetylase inhibitor SAHA [79]. This compound was also overproduced by the deletion of the *stc2* gene encoding an unknown sesquiterpene cyclase. Its structure and absolute configuration were determined by extensive spectroscopic NMR and HRESIMS studies and electronic circular dichroism calculations. Its biosynthesis was studied by feeding ^2^H and ^13^C isotopically labelled precursors to the *B. cinerea Δstc2* mutant. The results were consistent with a mixed biosynthesis of botrycineric acid (**57**) as outlined in Figure 17, according to which *B. cinerea* biosynthesises α-ketoisocaproate and phenylpyruvate via the l-leucine and l-phenylalanine biosynthetic pathways. The reduction of α-ketoisocaproate to the ketoenol derivative and condensation with phenylpyruvate derivative leads to intermediate **58**. Subsequent intramolecular reactions, by nucleophilic attack of the hydroxyl group to the carbonyl group of the α-ketoisocaproate moiety, yields the intermediate cyclic hemiketal **59** which is oxidised to botrycinereic acid (**57**) (Figure 17) [79].

Lastly, a high redundancy of phytotoxic compounds contributing to the necrotrophic pathogenesis of *B. cinerea* has been revealed and reported using multiple knock-out mutants [80]. To comprehensively evaluate the contributions of most of the currently known plant-cell death-inducing proteins (CDIPs) and metabolites for necrotrophic infection, an optimised CRISPR/Cas9 protocol was established. Comparative analysis of mutants confirmed significant roles played by two polygalacturonases (PG1, PG2) and the phytotoxic metabolites botrydial (**1**) and botcinins for infection but revealed no, or only weak, effects from the deletion of the other CDIPs.

This was the first systematic study of the functional redundancy of fungal virulence factors and demonstrates that *B*. *cinerea* releases a highly redundant cocktail of proteins leading to the necrotrophic infection of a wide variety of host plants [80].

Moreover, the total synthesis of (+)-cinereain (**13**) (Figure 1), a fungal cyclotripeptide featuring a complex heterocyclic core isolated from *B. cinerea* in 1988 by Cutler et al. [25] featuring interesting plant growth regulating properties, has been achieved in a convergent manner [81], confirming the structure of **13** [25].

## 3. *Botryotinia fuckeliana* (de Bary) Wetzel (The Teleomorph Stage)

There are far fewer studies related to the secondary metabolism of *Botryotinia fuckeliana*, the *B. cinerea* teleomorph, than to its anamorphic form. The natural products obtained from this microorganism are limited to a few strains, specifically an endophytic strain and two marine strains [82,83,84,85,86]. Metabolic production of the fungi indicates the presence of compounds from terpenoids (sesquiterpenes and diterpenes), dipeptides and some hybrid compound classes. An important characteristic of the metabolic profile discovered is that it is different from the profile found in its anamorphic form. To date, there are no reports on the isolation of sesquiterpenes with a botryane skeleton (**1**,**2**), and only botcinin-type polyketides (**3**-**6**) [16,17] but no botrylactones (**8**) [21] (Figure 1) have been reported.

The first study of the secondary metabolism of the genus *Botryotinia* dates back to 2012. In that work, Kim et al. [82] obtained six botcinins (Figure 18) from the strain *Botryotinia* sp. SF-5275, isolated from Korean seaweed, two known compounds isolated from the fungus fermentation extract, botcinin A (**5**) and botcinin B (**6**), and four other compounds formed from the instability of 5 and 6 in methanolic solution. Known 3-*O*-acetylbotcinic acid methyl ester (**60**) and botcinin D (**62**) were derived from botcinin A (**5**) [16,82], and new compounds 3-*O*-acetylbotcineric acid methyl ester (**61**) and botcinin G (**63**) were obtained from modification by botcinin B (**6**) [16,82]. These observations indicated that a slow methanolysis of the hexahydropyrone moiety and elimination of acetic acid had occurred in compounds **5** and **6**.

The next study on *B. fuckeliana* secondary metabolism was conducted by Lin et al. [83] from the endophytic strain *B. fuckeliana* A-S-3 associated to *Ajuga decubens* Thunb, a plant of the Labiatae family used in traditional Chinese medicine [87]. Extracts of this strain were cytotoxic against several human cancer cells. Chemical studies resulted in the isolation of three sesquiterpenes with an ent-eudesmane skeleton and three cytochalasines (Figure 18): the new sesquiterpenes 1-keto-4α,15-epoxyeudesm-11-ol (**64**) and ent-4(15)-eudesmen-5,6-ol-1-one (**65**), and the known compounds ent-4(15)-eudesmen-11-ol-1-one (**66**), phenochalasin B (**67**), [12]-cytochalasin (**68**) and the [1,3]dioxacyclotridecin (**69**). Cytochalasans are part of a diversified group of fungal polyketide synthase, non-ribosomal peptide synthetase (PKS-NRPS) hybrid metabolites which have attracted much interest due to their wide range of biological activities [88].

All metabolites were tested for their cytotoxic activities against four human cancer cell lines (SMMC-7721 human hepatocellular carcinoma cell line, A549 human lung adenocarcinoma cell line, HepG2 hepatocellular carcinoma cell line and MCF-7 human breast adenocarcinoma cell line). Phenochalasin B (**67**) and [12]-cytochalasin (**68**) exhibited strong cytotoxicity, significantly inducing apoptosis in the human hepatocellular carcinoma cell line (HepG2).

Later, in 2019, Niu et al., carried out the chemical study of the marine strain *B. fuckeliana* MCCC 03A00494, isolated from deep-sea water in the Western Pacific Ocean, which produced 71 new (**70**-**140**) and eight known metabolites (**141**-**148**) (Figure 19 and Figure 20) [85]. All compounds obtained were aphidicolins, tetracyclic diterpenes with a 6/6/5/6 ring system which are potential nuclear DNA replication inhibitors in eukaryotic cells and in some viruses and have, therefore, been thoroughly studied for the treatment of cancer [89,90]. Among the isolated compounds, 12 still need to have their stereochemistry fully defined. Compounds **134**-**140** are rare norditerpenoids reported in this work for the first time.

Continuing with the study of the marine strain *B. fuckeliana* MCCC 03A00494, Niu et al. [84] isolated a new pimarane diterpene featuring a 6/6/6 tricycle system called botriopimarane (**149**) and ten known products (Figure 21). Pimarane derivatives are frequently produced by plants and fungi, and rarely by bacteria or marine organisms [91]. Bisabolane-type phenolic sesquiterpenes (**93-95**) and two derivatives from cyclonerodiol (**153** and **154**) were also obtained. Bisabolane-type phenolics have been isolated from several marine organisms such as the fungus *Aspergillus* sp., corals and sponges [92]. The same characteristics are noted for cyclonerodiol derivatives which have already been found in *Aschotria* sp., *Trichothecium roseumm* and *Trichoderma harzian*um marine fungi [93,94,95].

Additionally, four cyclodipeptides and the compound 5α,8α-epidioxyergosta-6,22-dien-3β-ol (**155**) were isolated in the same study. The cyclodipeptides, or 2,5-diketopiperazine [96], are one of the simplest peptidic derivatives found in nature and have already been observed in plants, microorganisms and animals [97]. The cyclodipeptides isolated from *B. fuckeliana* were: cyclo-(L-Leu-L-Trp) (**156**), cyclo-(L-Trp-Gly) (**157**), cyclo-(L-Phe-L-Leu) (**158**), cyclo-(L-Leu-L-Tyr) (**159**).

The latest study to date on the marine strain *B. fuckeliana* MCCC 03A00494 [86] reported the isolation of eight tetracyclic diterpenes, botryotins A-H (**160-167**) (Figure 21). The new metabolites had skeletons representing three new 6/6/5/5, 6/6/5/6 and 6/6/6/5 tetracyclic ring systems. Botryotins A-F (**160-165**) were derived from aphidicolins by means of a Wagner–Meerwein rearrangement in the D ring, while compounds **166** and **167** featured unprecedented configurations. Additionally, the anti-allergic and anti-proliferative activities of botryotins were evaluated, noting that all compounds were inactive except botryotin A (**160**), which exhibited moderate anti-allergic activity compared to the standard loratadin.

## 4. Conclusions

This review is a comprehensive account of the genes involved in the biosynthesis of secondary metabolites of the plant pathogenic fungus *B. cinerea* and of all the metabolites isolated from this fungus from January 2015 to October 2022. In this interval, many metabolites have been isolated and characterised from both morphic forms of the phytopathogenic fungus: approximately forty metabolites were characterised from the anamorphic stage (*B. cinerea*) and, interestingly, a higher number of metabolites, approximately 110, were isolated from the teleomorph stage (*Botryotinia fuckeliana*).

The recent genome resequencing of *B. cinerea* strains, namely B05.10 and T4, has revealed 44 genes encoding key enzymes (KE) involved in the biosynthetic pathways of secondary metabolites of the fungus. These ranged from STCs and DTCs (sesquiterpene and diterpene cyclases), to PKSs and NRPS (polyketide synthases and non-ribosomal peptide synthetases) and some dimethylallyltryptophan synthases (DMATS). Most of these KE-encoding genes are co-localised with other enzyme-encoding genes that probably contribute to the biosynthesis of the same SM by modifications of the original skeleton. Taking into account the number of these gene clusters, *B. cinerea* may be able to produce approximately 40 different families of compounds; however, only a small number of metabolite families has been characterised thus far.

An important factor hampering the identification of fungal SMs is the fact that some of the biosynthesis gene clusters are silent under standard cultivation conditions. In recent years, the field of genome mining has emerged and many secondary metabolites (SMs) have been characterised from fungi sequenced using genomics-guided approaches [98].

In this review, new metabolites which have not been isolated from the anamorphic stage of *B. cinerea* were isolated from *Botryotinia fuckeliana* and were characterised. The diterpene cyclases DTC1, DTC2, DTC3 and PAX1 of the *B. cinerea* fungus remain uncharacterised. However, while no diterpenes have been detected in the broths of *B. cinerea*, approximately 90 diterpenes, 71 aphidicolins, 8 nor-diterpenes, 1 botry-pimarane and a further 8 diterpenes named botryotins A-H have been reported from a strain of *Botrytinia fuckeliana* from the Western Pacific Ocean and from some endophytic strains.

Clearly, the production of SMs depends on environmental signals that could be either abiotic or biotic. Both the improvement of culture conditions and a better knowledge of the regulation of secondary metabolism would be helpful in this regard (reviewed in [99,100]).

Lastly, further study of *B. cinerea* secondary metabolism may provide a useful starting point for the identification of new biological targets to control this crop-devastating fungus. This wealth of knowledge may contribute to the design of selective and rational structure-based fungicides [61].

## Figures and Tables

**Figure 1 plants-12-00553-f001:**
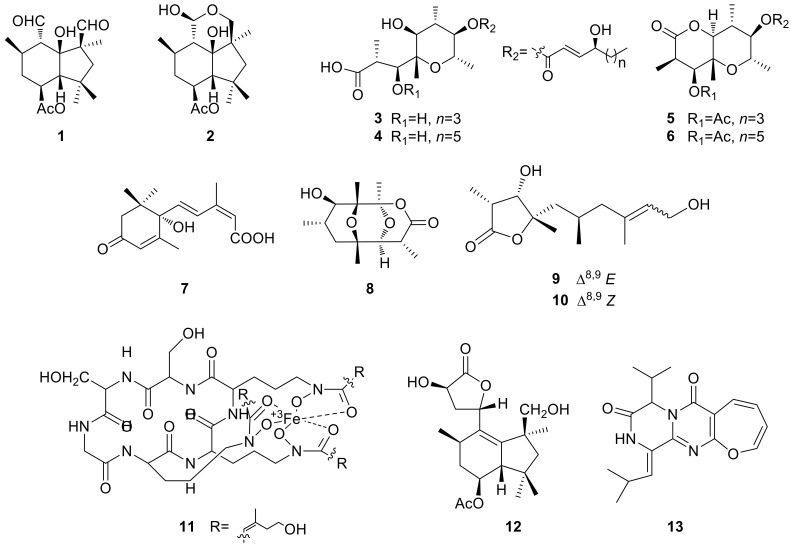
Some of the metabolites isolated from *B. cinerea*.

**Figure 2 plants-12-00553-f002:**
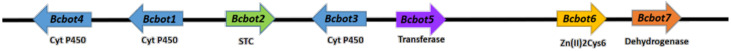
The botrydial gene cluster in the *B. cinerea* strain B05.10.

**Figure 3 plants-12-00553-f003:**
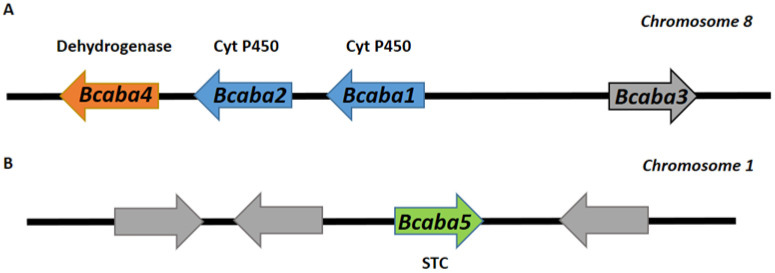
(**A**) The ABA gene cluster (**B**) the *Bcsct5/Bcaba5* gene locus in the genomes of *B. cinerea* strain B05.10.

**Figure 4 plants-12-00553-f004:**
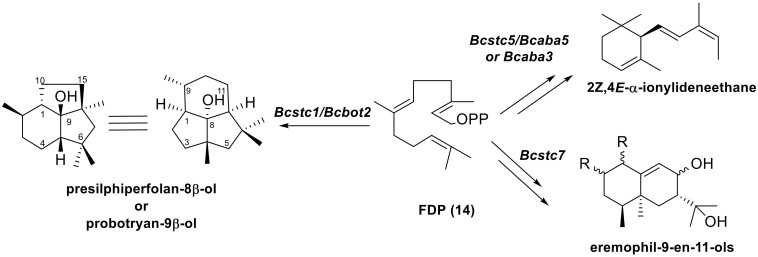
Cyclisation of FDP (**14**) by *Bcstc1, Bcstc5* or *Bcaba3* and *Bcstc7*.

**Figure 5 plants-12-00553-f005:**
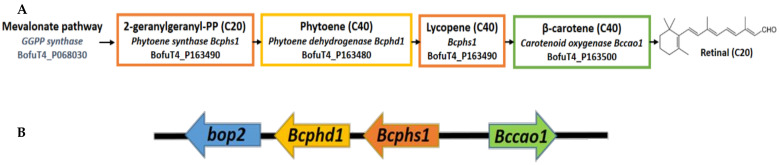
(**A**) Proposed pathway for carotenoid biosynthesis in *B. cinerea*. Genetic makeup may allow for the production of retinal as an end product of a branched pathway. (**B**) The genes required for retinal biosynthesis are physically linked with the opsin-encoding *bop2* in *B. cinerea*.

**Figure 6 plants-12-00553-f006:**
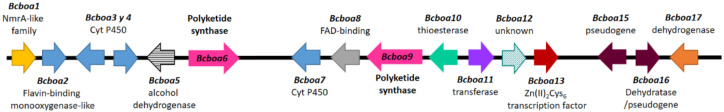
BOA gene cluster in the genomes of *B. cinerea* strain B05.10.

**Figure 7 plants-12-00553-f007:**
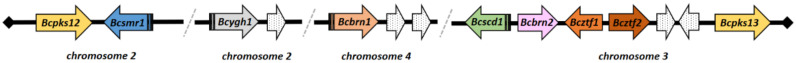
Melanogenic genes from *B. cinerea*.

**Figure 8 plants-12-00553-f008:**
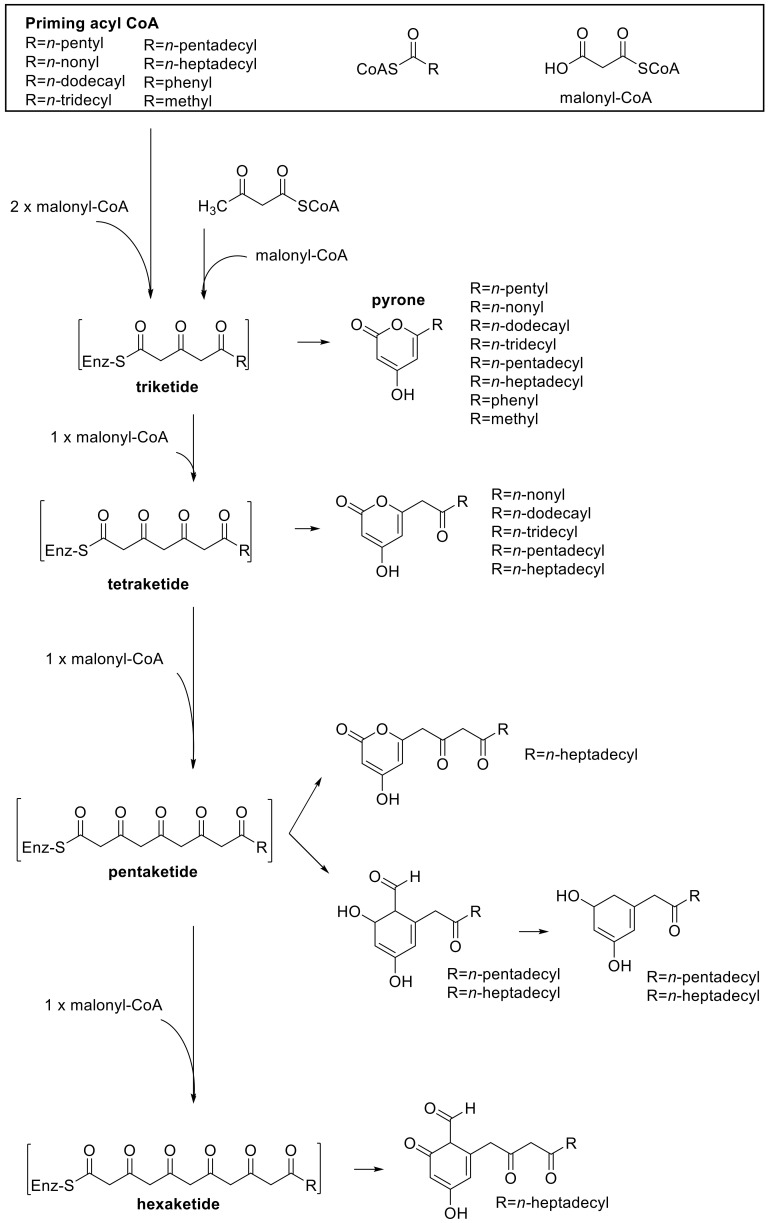
Summary of BPKS reactions with various acyl-CoA starter substrates.

**Figure 9 plants-12-00553-f009:**
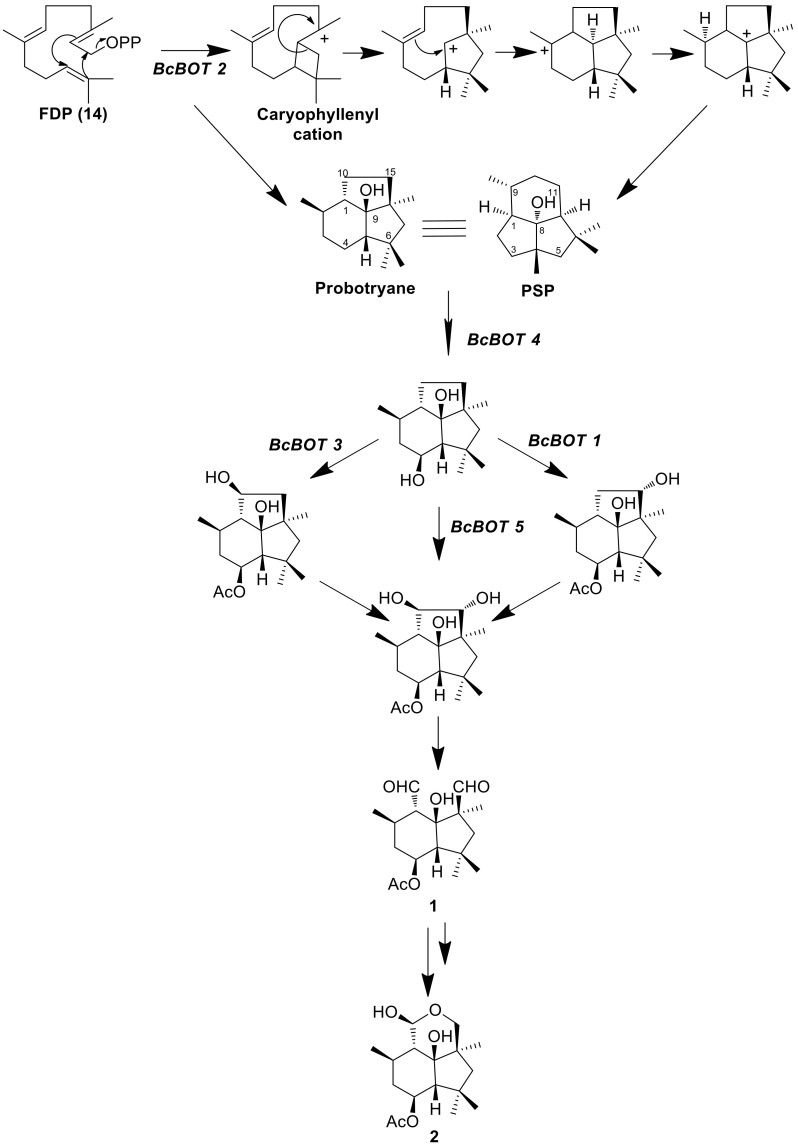
Biosynthetic pathway from FDP (**14**) to botrydial (**1**) and dihydrobotrydial (**2**).

**Figure 10 plants-12-00553-f010:**
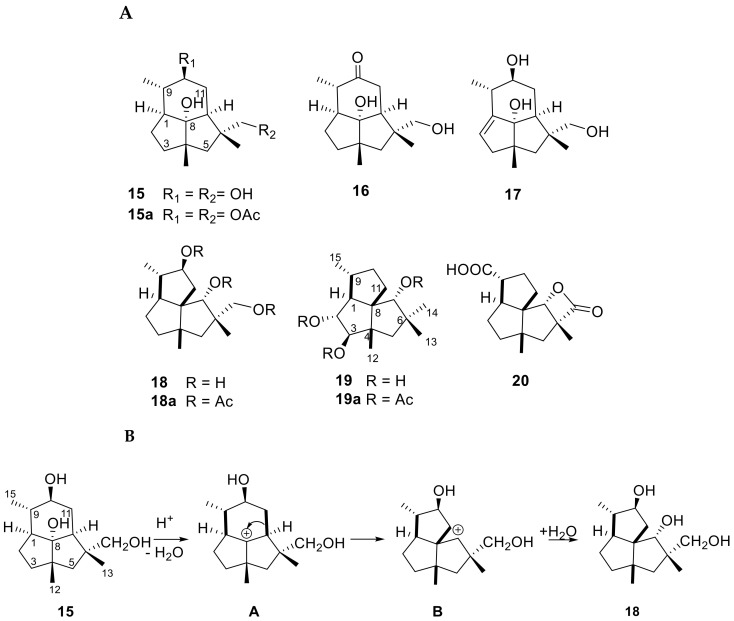
(**A**) Presilphiperfolanes and cameroonanes isolated from the *bcbot-4* knock-out mutant fermentation. (**B**) Rearrangement of the presilphiperfolane to cameroonane skeleton.

**Figure 11 plants-12-00553-f011:**
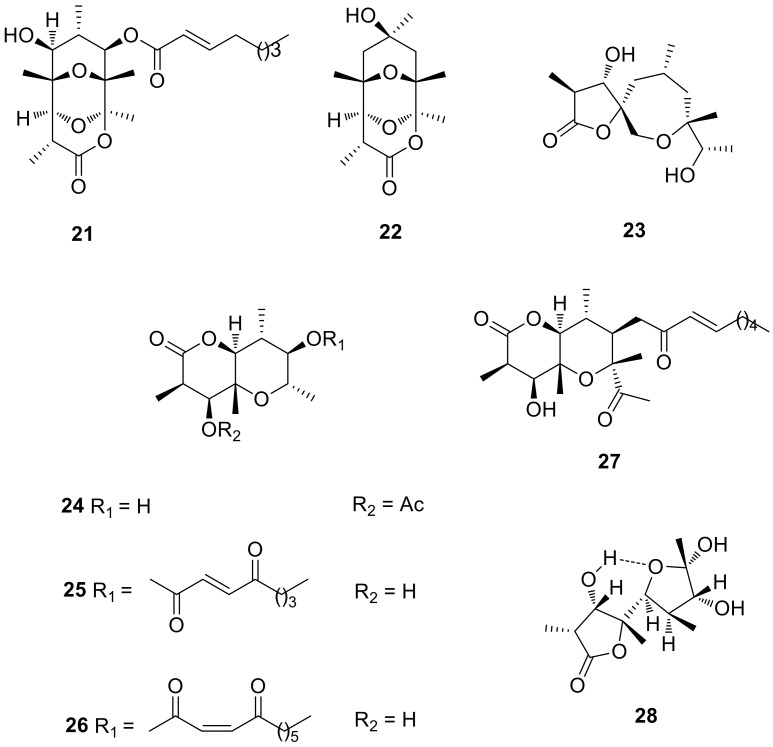
New polyketides isolated from *Bcbot4* knock-out mutant fermentations.

**Figure 12 plants-12-00553-f012:**
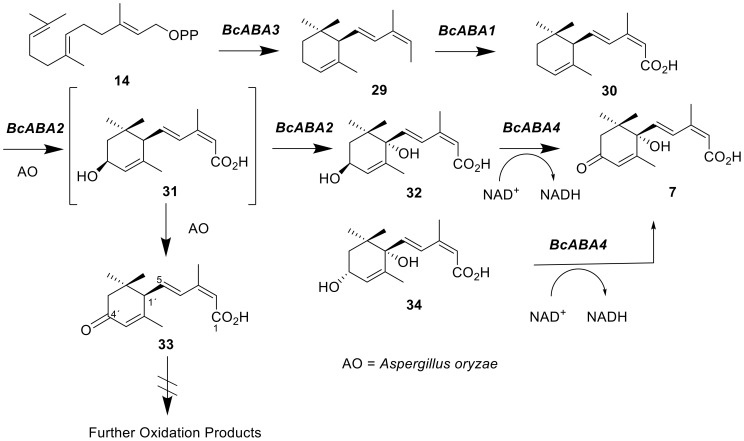
Biosynthetic pathway of **7** in *B. cinerea*.

**Figure 13 plants-12-00553-f013:**
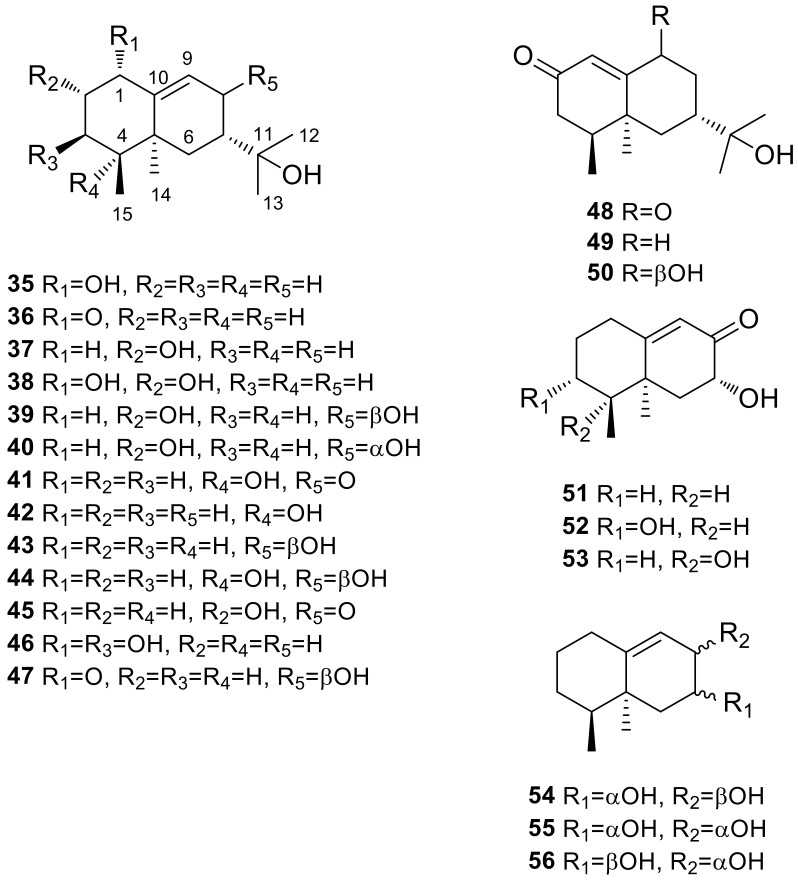
Eremophilene derivatives isolated from *B. cinerea*.

**Figure 14 plants-12-00553-f014:**
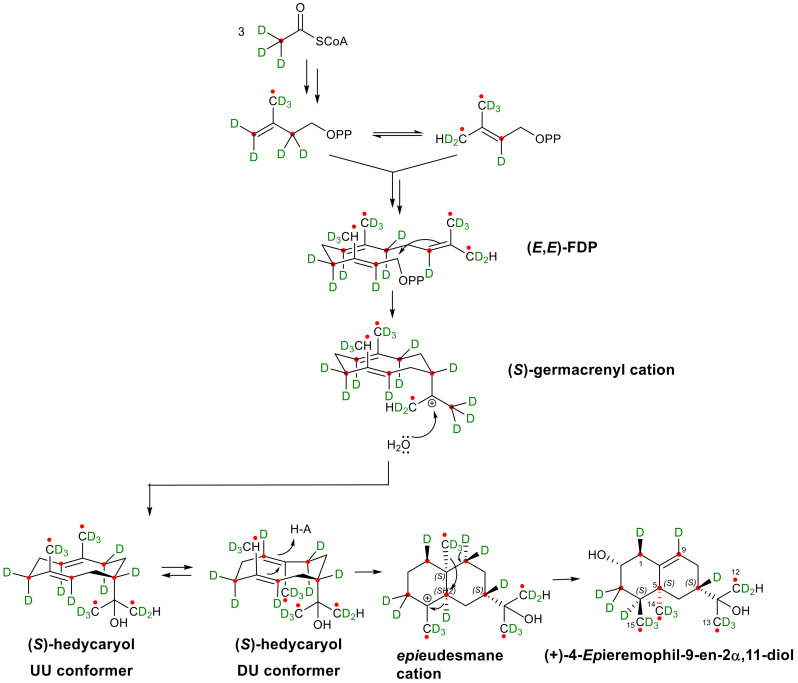
Proposed biosynthetic route to (*+*)-4-*epi*-eremophilenols.

**Figure 15 plants-12-00553-f015:**
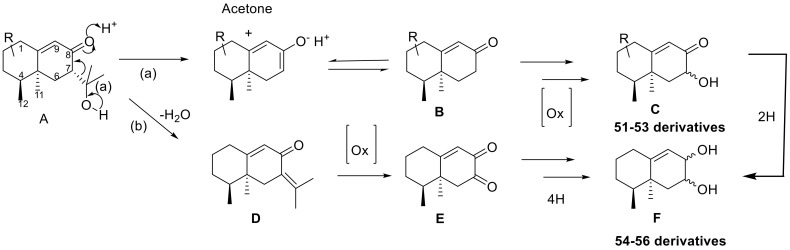
Biosynthetic pathway proposed for 11,12,13-tri-*nor*-eremophilenes.

**Figure 16 plants-12-00553-f016:**
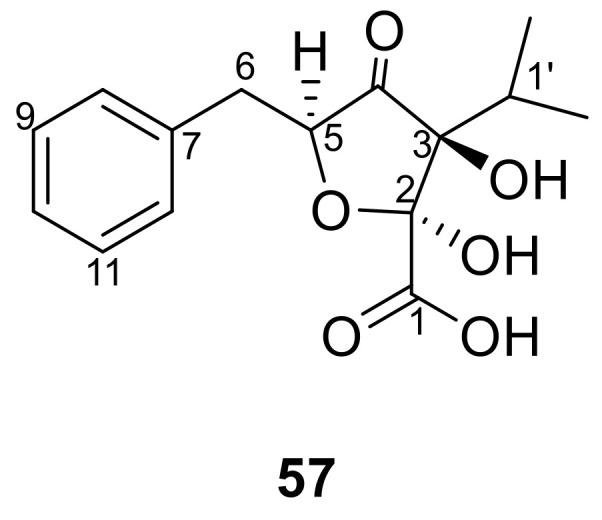
Botrycinereic acid.

**Figure 17 plants-12-00553-f017:**
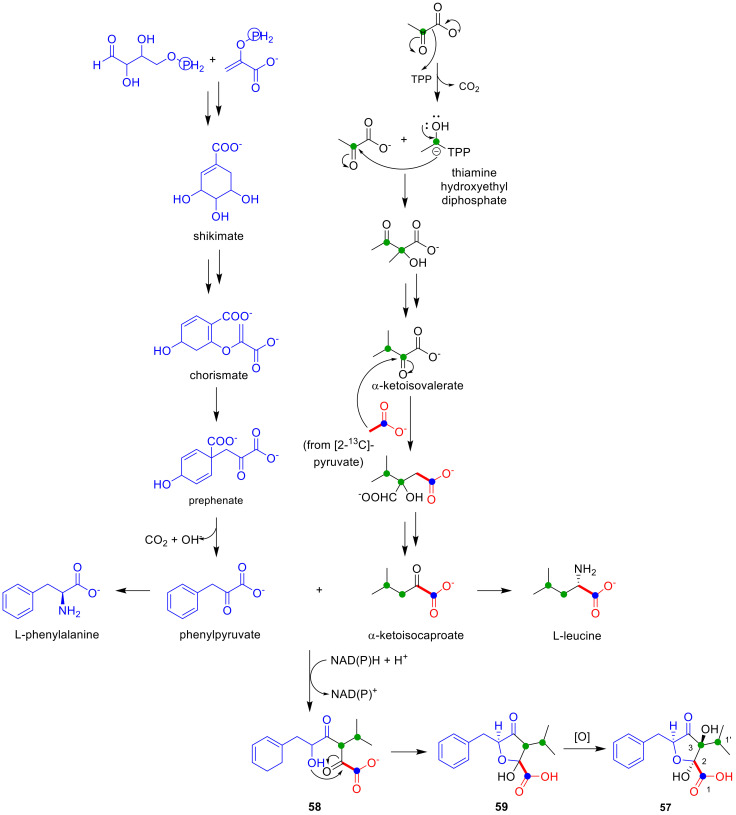
Proposed biosynthetic pathway to botrycinereic acid (**57**).

**Figure 18 plants-12-00553-f018:**
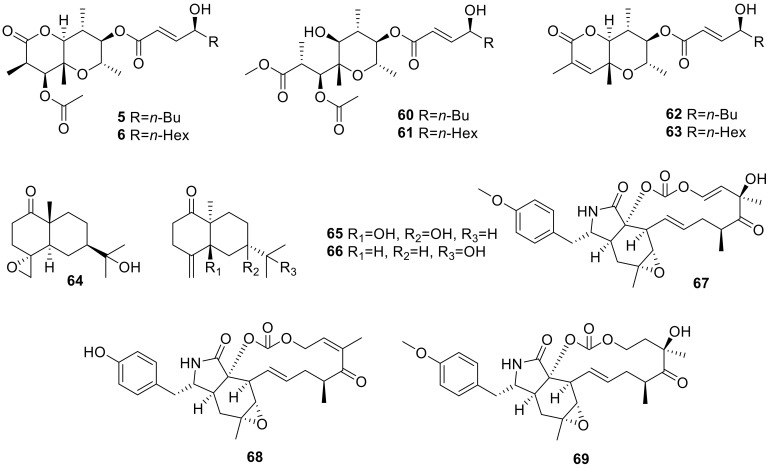
Structures of compounds **5**,**6**,**60-69**.

**Figure 19 plants-12-00553-f019:**
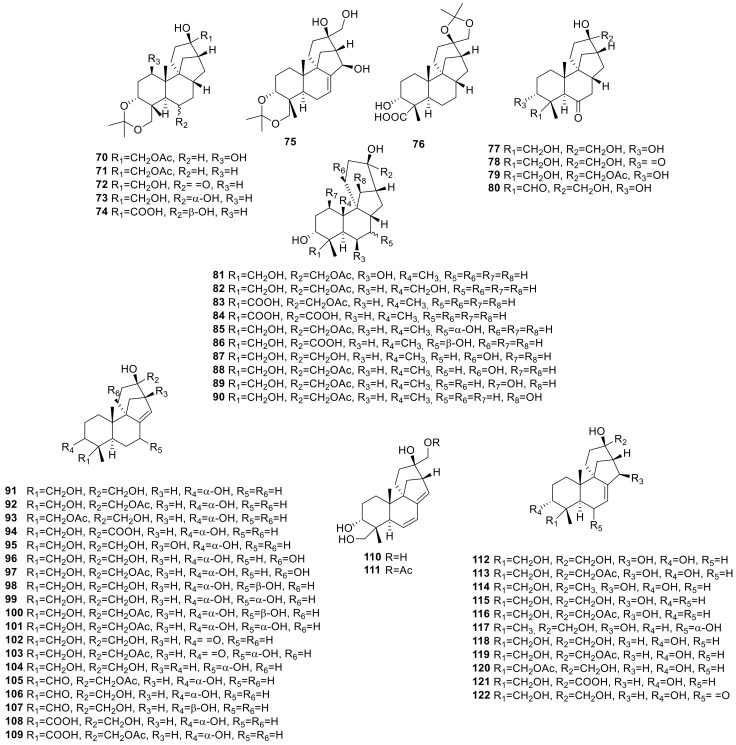
Structures of aphidicolin derivatives **70**-**122**.

**Figure 20 plants-12-00553-f020:**
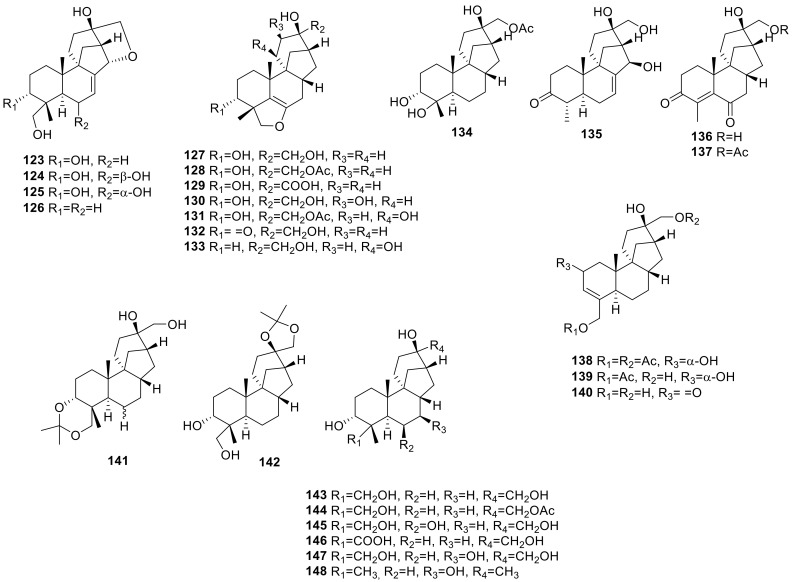
Structure of other diterpenoid derivatives **123**-**148**.

**Figure 21 plants-12-00553-f021:**
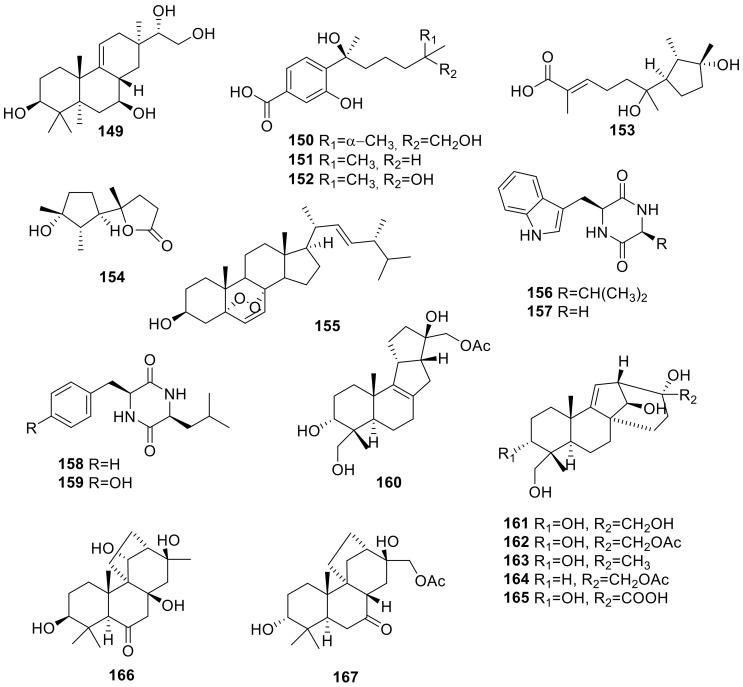
Structure of compounds **149-167**.

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
