# Peer review of "From Genes to Molecules, Secondary Metabolism in Botrytis cinerea: New Insights into Anamorphic and Teleomorphic Stages"

_plants, 2023, doi:10.3390/plants12030553_

Round 1

Reviewer 1 Report

This paper reviews the identified secondary metabolites and corresponding biosynthetic pathways of Botrytis cinerea (the grey mould fungus) in great detail. The authors find that there are still some papers using the teleomorph name, Botryotinia fuckeliana, and the new metabolites isolated from these strains are also introduced. This paper could be a nice reference for researchers who are interested in synthetic biology and botrytis control.

P13, "Both PLC and DGK inhibition reduce ROS production sparked by botrydial (1). This shows that ROS production is upstream of PLC/DGK."

The express is wrong. PLC/DGK should be upstream of ROS production instead, according to the context.

Author Response

- Following the suggestions of this referee, we have corrected the indicated sentence in the page 13, (see marked manuscript).

Reviewer 2 Report

The review is extensive and in my opinion, gives a good overview of the topics. However, I think that the authors should address some writing problems.

Your language is a little tricky and hard to understand in many parts of the paper. So, I suggest just rephrasing some sentences but detailed comments can be found in the PDF

Author Response

- Following the interesting comments of referee 2, in order to improve the manuscript, we are accepted your suggestions about different sentences throughout the manuscript. (see marked manuscript in yellow)

Additionally, the manuscript has been revised by an English native translator